# *Amaryllidaceae*, *Lycopodiaceae* Alkaloids and Coumarins—A Comparative Assessment of Safety and Pharmacological Activity

**DOI:** 10.3390/jcm11154291

**Published:** 2022-07-23

**Authors:** Mariola Herbet, Jarosław Widelski, Iwona Piątkowska-Chmiel, Kamil Pawłowski, Aleksandra Dymek, Tomasz Mroczek

**Affiliations:** 1Department of Toxicology, Faculty of Pharmacy, Medical University of Lublin, 20-090 Lublin, Poland; iwona.piatkowska.chmiel@umlub.pl (I.P.-C.); kamil.pawlowski@umlub.pl (K.P.); 2Department of Pharmacognosy with the Medicinal Plant Garden, Medical University of Lublin, 20-093 Lublin, Poland; jaroslaw.widelski@umlub.pl; 3Department of Chemistry of Natural Products, Medical University of Lublin, 20-093 Lublin, Poland; aleksandra.dymek91@interia.pl (A.D.); tmroczek@pharmacognosy.org (T.M.)

**Keywords:** Alzheimer’s disease, *Amaryllidaceae* alkaloids, *Lycopodiaceae* alkaloids, coumarins, acetylcholinesterase inhibitors, inflammation

## Abstract

The study aimed to evaluate the safety and pharmacological activity *Amaryllidaceae*, *Lycopodiaceae* alkaloids and coumarins obtained from *Narcissus triandrus* L., *Lycopodium clavatum* L., *Lycopodium annotinum* L., *Huperzia selago* L. and *Angelica dahurica* (Hoffm.) Benth. & Hook.f. ex Franch. & Sav. In the in vivo studies. The influence of the tested compounds on the central nervous system of rats was assessed in behavioral tests (locomotor activity, Y-maze, passive avoidance). In order to investigate the mechanisms of action, biochemical determinations were performed (AChE activity, BChE activity, IL-1β, IL-6 concentration). In order to assess safety, the concentrations of AST, ALT, GGT and urea and creatinine were determined. The results of the conducted studies indicate a high safety profile of the tested compounds. Behavioral tests showed that they significantly improved rodent memory in a passive avoidance test. The results of biochemical studies showed that by reducing the activity of AChE and BChE and lowering the concentration of IL-1β and IL-6, the coumarin-rich *Angelica dahurica* extract shows the most promising potential for future therapeutic AD strategies.

## 1. Introduction

One of the most common neurodegenerative diseases and the most common cause of cognitive impairment in the elderly is Alzheimer’s disease (AD). The number of AD patients continues to grow; currently, more than 46 million people worldwide suffer from the disease. According to estimates by the World Health Organization (WHO), the predicted prevalence of this disease in the global population will quadruple in the next decades, reaching 114 million patients by 2050 [1]. Multiple mechanisms are involved in the pathogenesis of AD. There are several hypotheses relating to its pathogenesis, including, among others, the cholinergic hypothesis, amyloid cascade hypothesis, inflammation hypothesis and genetic hypothesis.

A key role in the development of AD is played by dysfunction of the cholinergic system, which includes cholinergic neurons, neurotransmitters and their receptors [2,3]. Acetylcholine (ACh) is a neurotransmitter used by cholinergic neurons and is involved in the processes of attention, learning, memory, stress response, wakefulness and sleep, and sensory information [4,5,6,7,8,9]. Cholinergic disorders result from the loss of cholinergic neurons in the basal forebrain and hippocampus, which reduces cognitive ability [2,3]. According to this hypothesis, the cholinergic innervation is disturbed in the early stages of AD, and Ch4 neurons are especially susceptible to degeneration. AChE overactivity is observed, which leads to a decrease in ACh concentration and, in turn, causes degeneration of the cholinergic system [10,11,12,13]. It was shown that cells with increased expression of this enzyme on their surface undergo apoptosis more quickly [14,15]. Thus, damage to cholinergic neurons was considered a critical pathological change that correlated with cognitive impairment in AD [9].

Neuritis is also a hallmark of AD. According to the conducted research, the inflammatory response plays a key role in the process of neurodegeneration during the progression of AD. The pathways associated with microglia were recognized as crucial in the risk and pathogenesis of AD, as confirmed by new genetic studies [16,17,18,19,20]. Synaptic plasticity processes are the basic cellular basis of learning and memory [9,21]. Reactive microglia and astrocytes surround the amyloid plaques and secrete numerous pro-inflammatory cytokines, which are considered early and major drivers in the development of AD [9]. Both genetic and biochemical studies have shown overexpression of the pro-inflammatory cytokine interleukin β (IL-β) in patients with the disease compared to the control group [22]. An association was also demonstrated between the process of neurodegeneration and inflammatory biomarkers such as interleukins, interferon (IFN)-γ, tumor necrosis factor (TNF)-α, transforming growth factor (TGF)-β and acute phase reagent protein c (CRP) [23,24,25,26,27].

ACh, the decline in which ACh levels in cholinergic neurons are characteristic of AD, is broken down by acetylcholinesterase (AChE) and butyrylcholinesterase (BChE). Therefore, inhibition of its action by drugs belonging to the group of ACHE inhibitors leads to the improvement of cognitive functions. The American Food and Drug Administration (FDA) certified AChE inhibitors as the first drugs to treat AD [28]. Therapy with the above drugs relies partially on the symptoms of cognitive impairment by increasing neurotransmission in the brain. However, these drugs do not stop the progression of the disease and have many side effects. In turn, nonsteroidal anti-inflammatory drugs (NSAIDs) have not provided sufficient clinical benefit since the relationship between innate immunity and AD pathogenesis is complex, and the immune response may be harmful or beneficial depending on the context [29,30,31]. Therefore, intensive attempts are made to search for new, strong AChE inhibitors with higher efficacy and higher safety that could be used in the treatment of AD. 

Natural substances are a very interesting target of these strategies [32]. The results obtained from recent studies clearly indicate that Narcissus ‘Hawera’ among other plants from the genus Narcissus and, generally, the *Amaryllidaceae* family is a very rich source of bioactive alkaloids [33]. These compounds, such as the well-known galantamine isolated from *Narcissus bulbs*, have the potent and reversible ability to inhibit AChE. Moreover, galantamine is currently being used in attempts to treat neurological disorders characterized by memory impairment, cognitive dysfunction, behavioral disorders and deficits in activities of daily living. Over time, researchers have proven that the alkaloid huperzine A (HupA), isolated from the *Huperzia species* (the *Lycopodiaceae* family), penetrates better the blood–brain barrier, has higher bioavailability and is less toxic compared to known AChE inhibitors (e.g., galantamine). Furthermore, recent studies in *Huperzia*, as well as *Lycopodium species*, confirmed the presence of novel potent AChE inhibitors in vitro assays [34]. *Angelica dahurica* (Hoffm.) Benth. & Hook.f. ex Franch. & Sav. root is an abundant source of phytochemicals, among them compounds belonging to furano- and piranocoumarins [35]. Traditional Chinese Medicine (TCM) is used as a drug (called Bai Zhi) for the treatment of neurogenic headaches, anxiety, fever, skin rashes, wounds, rheumatism and toothaches [36]. Many data suggest that these three coumarins have high BBB (blood–brain barrier) permeability and have a pharmacokinetic potential for the treatment of central nervous system diseases. 

Preliminary studies on the search for strong ACHE inhibitors have shown their effectiveness and safety in in vitro tests, while there are no data in the literature confirming their safety and pharmacological activity in the in vivo study. In connection with the above, the study aimed to evaluate the safety and pharmacological activity of plant extracts obtained from *Narcissus triandrus* L., *Lycopodium clavatum* L., *Lycopodium annotinum* L., *Huperzia selago* L. and *Angelica dahurica* (Hoffm.) Benth. & Hook.f. ex Franch. & Sav. in *in vivo* studies. The effect of extracts containing *Amaryllidaceae*, *Lycopodiaceae* alkaloids and coumarins on the central nervous system of rats was assessed in behavioral tests. Additionally, to investigate the mechanisms of the extracts’ action, biochemical determinations were performed. Moreover, in order to assess the safety of the tested extracts, biochemical parameters indicating the function of the liver and kidneys in rats were determined. Our work is the first in which the above extracts, isolated and standardized using state-of-the-art techniques, were evaluated for safety and effectiveness in rodent studies.

## 2. Materials and Methods

### 2.1. Plant Materials

The plant materials used for the research were whole plants of five different species or their parts, oven-dried and powdered. Three of them were club moss species of the *Lycopodiaceae family: Lycopodium clavatum* L., *Lycopodium annotinum* L. and *Huperzia selago* L. and the others were species of *Narcissus triandrus* L. c.v. ‘Hawera’ (*Amaryllidaceae* family) and *Angelica dahurica* (Hoffm.) Benth. & Hook.f. ex Franch. & Sav. (*Apiaceae* family). Specimens of whole plants of the *Lycopodiaceae* family were collected in southern Poland and western Ukraine. The plant material of *Lycopodium annotinum* was of Polish origin, and it was collected in the Tatras. The other two species were harvested in the Oblast Rivne, Ukraine: *Huperzia selago* from the Volodymyrets region and Lycopodium clavatum from the Sarny region. In the case of *Narcissus* c.v. ‘Hawera’, bulbs with appropriate authenticity certificates purchased from Florexpol (Lublin, Poland) were used for the study. All plant species were authenticated by Prof. Mroczek and deposited in the Department of Chemistry of Natural Products, the Medical University of Lublin in Poland. *Angelica dahurica* radix was bought from Planetherbs company (planetherbs.pl) and was obtained from China.

### 2.2. Plant Materials Extraction

Plant materials were extracted by pressurized liquid extraction (PLE) using Dionex ASE 100 extractor (Sunnyvale, CA, USA) at elevated temperature and pressure. The ground dried plant materials were accurately weighed (in triplicate): 0.5 g of powder from each of the three *Lycopodiaceae species*, 1 g of powdered bulbs of *Narcissus* c.v. ‘Hawera’ and 10 g of powdered *Angelica dahurica*. The weighted and pulverized raw materials were placed in a 10 mL stainless steel extraction cell and extracted twice with pure methanol as a solvent. Additionally, 2 g of celite (a deactivating adsorbent), used to maintain a constant temperature and seal the cell, were added to the cell with the powdered *Narcissus* c.v. ’Hawera’. The extraction procedures and their conditions for these species were carefully tested and optimized, and the results were described in detail [37,38]. Accordingly, extraction of *Narcissus* c.v. ’Hawera’ species were carried out at 140 °C and 100 bar, while three *Lycopodium* species and *Angelica dahurica* were extracted at 80 °C and 100 bar. The operating conditions of the Dionex ASE 100 extractor (Sunnyvale, CA, USA) were set as follows: static time 10 min, flush volume 60%, number of extraction cycles 3. The concentrated methanolic extracts were transferred into 100 mL volumetric flasks and diluted to the mark with the solvent. This resulted in 5 extracts with the following concentrations: (1 g/100 mL for each of the three *Lycopodiaceae* species, 2 g/100 mL *Narcissus* c.v. *Hawera* extract and 20/100 mL extract of *Angelica dahurica*), which were further analyzed.

### 2.3. Animals

The experiments were carried out on 42 migratory rats, Wistar Han herd, male, 8 weeks old and sexually mature (190–230 g), purchased from a licensed breeder (Center for Experimental Medicine (EMC), Medical University of Lublin, Poland (077-EMC number in Lublin in the Register of Breeders kept by the Minister of Science and Higher Education (Poland)). It should be mentioned that we selected male offspring because we wanted to minimize the effect of alterations of sexual hormones on the results of behavioral tests. The animals were kept in standard conditions following the Regulation of the Minister of Agriculture and Rural Development of 14 December 2016 on the minimum requirements to be met by the center and the minimum requirements for the care of animals kept in the center (Journal of Laws, item 2139). Throughout the experiment, rats were kept in designated rooms in such conditions (temperature 22 ± 2 °C, humidity 50–55%, 15 air changes per hour, light cycle 12/12 h).

The animals were randomly divided into 6 experimental groups; each group consisted of 7 animals. Rats were divided into 6 groups and kept in home Plexiglas cages, with 2 animals each. Food and water were freely available to rats throughout the study, except when the tests were performed. Body weight was monitored once a little. The animals were used after 7 days of acclimatization to the laboratory conditions. During the 7-day laboratory adaptation period, rodents were habituated to human touch and smell. The rats were removed from the cage daily and touched for several minutes. These activities allow the animals’ stress to be minimized during the more invasive stages of the experimental procedure. Each experimental group consisted of 7 animals. The number of animals was estimated according to the requirements of statistical analyzes (Student’s *t*-test, ANOVA) and 3R (3R). Following this principle, the authors of the project decided to administer the tested substances in only one dose. The procedures involving rats and their care in all experiments of this study were approved by the Local Ethics Committee of the University of Life Sciences in Lublin (No. 75/2020) and were performed by the applicable European standards for experimental research on animal models (Act of January 15, 2015, on the protection of animals used for scientific or educational purposes; Directive 2010/63/EU of the European Parliament and of the Council of 22 September 2010 on the protection of animals used for scientific purposes). The experiment was carried out at EMC in Lublin. All activities were performed by qualified personnel, and the animals were under the constant care of a veterinarian; every effort was made to minimize suffering.

### 2.4. Drugs

The following substances were used in the study: Tween 80 (1% polyoxyethylene glycol sorbitan monooleate; POCH, Gliwice); 0.9% NaCl (Sigma-Aldrich, Poznan, Poland). All chemicals and reagents were of analytical grade.

### 2.5. Experimental Design

The animals were divided into six groups. Rats were assigned to the experimental groups according to weight as follows: (I)—rats receiving once a day for 28 days *aqua pro injectione* (through a gastric tube, per os); (II)—rats receiving once a day for 28 days the freeze-dried methanol extract from bulbs from the *Narcissus triandrus* L. cultivar at a dose of 20 mg/kg (per os); (III)—rats receiving once a day for 28 days a lyophilized methanol extract from the aerial parts of *Lycopodium clavatum* L. at the dose of 10 mg/kg (per os); (IV)—rats receiving once a day for 28 days a freeze-dried methanol extract from the aerial parts of *Lycopodium annotinum* L. at the dose of 10 mg/kg (per os)*;* (V)—rats receiving once a day for 28 days the lyophilized methanol extract from the aerial parts of *Huperzia selago* L. at the dose of 10 mg/kg (per os); (VI)—rats receiving once a day for 28 days lyophilized methanol extract from *Angelica dahurica* (sp.) fruit at a dose of 200 mg/kg (per os). Plant extracts were suspended in 1% Tween 80 in saline, and the solutions were prepared in this way and then were administered per os in a volume of 1 mL/100 g of the body weight. The doses of plant extracts were selected based on the literature data [39,40,41,42,43]. After 28 days of administration of the aqua pro injectione and extracts, the animals were subjected to behavioral tests assessing the effect on motor coordination, cognitive activity and memory capacity.

### 2.6. Behavioral Studies

#### 2.6.1. Locomotor Activity Test

Twenty-four hours after the last administration of plant extracts to the animals, a test was performed to assess the locomotor activity of the animals. The rats were placed individually (in the order in which they had previously received the tested extracts) in a special apparatus designed to measure locomotor activity. The camera is a transparent square cage (Porfex, Białystok, Poland) made of plexiglass with dimensions of 60 × 60 cm^2^. The walls of the cage are equipped with two rows of infrared photocells located 45 and 100 mm above the floor. During the test, the horizontal locomotor activity (distance traveled) undertaken by each rat for 15 min was recorded. The test was performed in a soundproofed experimental room lit with subdued light (red bulb). At the end of each measurement, animals were removed from their cages and returned to their housing cages. The measuring device was washed with an ethyl alcohol solution (10% *v*/*v*) to eliminate the olfactory sensations.

#### 2.6.2. Y Maze Test

The next day (48 h after the administration of the extracts), the Y-maze test was performed, which is used to assess the rodents’ readiness to discover new environments and to assess cognitive functions. The Y maze is a working memory task that requires rats to use the outer maze cues to navigate on identical inner arms. Rodents usually prefer to explore a new maze arm rather than return to a previously visited one. The maze consists of 3 opaque plastic arms (A, B, C) 35 cm long, 25 cm high and 10 cm wide, set at an angle of 120° to each other. Each animal was placed in the center of the maze, where it was free to explore its three arms. The number of arm entries and the number of triads were recorded to calculate the percent change. Entry occurs when all four limbs are in the shoulder. The test lasted 8 min; the sequence of the arms of the maze into which the animal entered was noted. On this basis, the physical activity (number of runs) and fresh spatial memory (% of logical changes) can be assessed. The percentage of spontaneous lesions was calculated using the formula: [(number of alternations)/(total number of arm entries −2)] × 100. If a rat scored significantly above 50% alternations (the chance level for choosing the unfamiliar arm), this indicates functional working memory. At the end of each measurement, the animals were removed from the maze and placed in their cages. To prevent errors in data analysis, the experimenter blindly ran the Y-maze test.

#### 2.6.3. Passive Avoidance Test

After another 24 h, a passive avoidance test was carried out to assess the influence of the tested extracts on the cognitive effects of rats. An apparatus consisting of two identical rooms with dimensions of 22 cm in length × 28 cm in width × 22 cm in height was used for the study. The first room was illuminated by fluorescent light (8W), and the second was equipped with a cover that provided darkness. The rooms were separated by guillotine doors. The floor in both rooms is made of metal bars 2 mm thick, spaced 1 cm apart. The experiment procedure consists of a training session (preliminary test, pre-test) and a proper test. Habituation of animals into the room was performed 24 h before the start of the training session. For this purpose, each animal was placed in the apparatus and left for 3 min to freely explore the apparatus and become acquainted with the new surroundings. In the training session, rats were placed individually in the lighted chamber and allowed to move freely in it. After a 30 s adaptation phase, the door was raised to allow animals to enter the dark room. Following their instinct, rodents quickly move to a dark room, which they perceive as safe. After the rat had passed into the dark chamber, the door was closed with all paws, and the animal’s feet were electrocuted (0.2 mA) for 2 s (aversive stimulus). During the training session, the time taken for the rat to enter the dark chamber (Tl1) was measured. After 24 h, the proper test was performed, which was analogous to the preliminary test, but without generating an aversive stimulus just after the animal passed into a dark room. During this test, the time taken to enter the dark chamber (Tl2) was recorded, and the STL was measured. The waiting time for the rat to enter the dark room was 300 s. After each test, the apparatus was rinsed with an ethyl alcohol solution (10% *v*/*v*) to eliminate the olfactory sensations. A high value of STL indicates a positive effect of the tested substances on learning and memory processes, while a low value indicates a disturbance in cognitive processes and short-term memory. The rats were euthanized using a method compliant with Polish and European regulations (decapitation).

### 2.7. Biochemical Studies

#### 2.7.1. Collection of Blood

After the behavioral tests, the rats were decapitated by experienced animal technicians with the appropriate certificates. The murine blood was collected into Eppendorf tubes and allowed to clot at room temperature. Then, the blood was centrifuged for 10 min at 5000× *g*, and serum was collected into polyethylene tubes and frozen at −25 °C. The ready-made diagnostic kits were used to determine the activity levels of aspartate aminotransferase (AST), alanine aminotransferase (ALT), gamma-glutamyl transferase (GGT), the concentrations of urea and creatinine: Liquick Cor-AST-60, Liquick Cor-ALT-60, Liquick Cor-GGT, Liquick Cor-UREA 120, Liquick Cor-CREATININE 60 (all: PZ Cormay S.A., Łomianki, Poland). The activity of AChE and BChE ELISA Kit for Acetylcholinesterase (ACHE) in the serum of rats was measured by ELISA Kit for ACHE and BChE (Cloud–Clone Corp., Katy, TX, USA). The concentration of IL-1β and IL-6 was measured by a ready-to-use sandwich enzyme immunoassay (ELISA) diagnostic kit dedicated to rats fluids (ELISA Kits for IL-1β, IL-6, Cloud–Clone Corp., Katy, TX, USA). All procedures were conducted according to the manufacturer’s instructions.

#### 2.7.2. Collection of Brains

After the behavioral tests, the rats were decapitated by experienced animal technicians with the appropriate certificates. The brain of rats was carefully removed immediately after the decapitation and immersed in cooled (2–8 °C) saline to remove blood. The brains were isolated and washed with a 20 µL injection solution and stored in a freezer at −80 °C for biochemical studies. The activity of AChE and BChE ELISA Kit for Acetylcholinesterase (ACHE) in the prefrontal cortex of rats was measured by ELISA Kit for AChE and BChE (Cloud–Clone Corp., Katy, TX, USA). The concentration of IL-1β and IL-6 was measured by a ready-to-use sandwich enzyme immunoassay (ELISA) diagnostic kit dedicated to rats’ tissues (ELISA Kits for IL-1β, IL-6, Cloud–Clone Corp., Katy, TX, USA). All procedures were conducted according to the manufacturer’s instructions.

### 2.8. Statistical Analysis

The data were presented as means ± standard error of the mean (SEM). In order to evaluate the effect of plant extract treatment on measured parameters, a one-way ANOVA followed by a post hoc Tukey’s test was used. The results were analyzed statistically using STATISTICA 12.0 application (StatSoft, Cracow, Poland). Differences were considered to be significant at *p* < 0.05.

## 3. Results

### 3.1. Body Weight of Rats

The results obtained for the body weight of rats are presented in Table 1. Our study showed no statistically significant changes in the body weight of the animals receiving the test extracts for 28 days compared to the control group.

### 3.2. Behavioral Studies

As shown in Figure 1, the physical activity in the locomotor activity task was not affected in rats subjected to all examined extract in comparison to the control group. In the second experiment, we examined the spontaneous alternation behavior and spatial memory in the Y-maze test (Figure 2a,b). In the experimental groups, except for the Lycopodium annotinum L. group, spontaneous alternation was above the chance level (50%). Our results revealed that the spontaneous alternation percentage of arm entries significantly increased in the Huperzia selago L. group as compared to the control group (F[2.64] = 3.822, *p* < 0.5). However, the total number of arm entries as compared to the control was significantly higher in the group of rats receiving *Narcissus triandrus* L. (F[2.64] = 2.913, *p* < 0.5). Learning and memory performance was evaluated in the passive avoidance test (Figure 3). On the 24 h retention trial of the passive avoidance test, using a maximum cut-off time of 300 s, the one-way ANOVA test showed a significant effect of extracts *Narcissus triandrus* L., *Lycopodium clavatum* L., *Huperzia selago* L. and *Angelica dahurica* on step-through latency (STL)—28 days pretreatment with examined extracts caused a significant increase in STL (F[2.64] = 4.151, *p* < 0.05, *p* < 0.01, *p* < 0.01, *p* < 0.01, respectively). The one-way ANOVA test showed no significant effect of *Lycopodium annotinum* L. on STL (*p* > 0.05).

### 3.3. Biochemical Studies

#### 3.3.1. AST, ALT, GGT Activity; Urea, Creatinine Concentrations

The results of blood tests determining the function of the liver and kidneys are presented in Table 2. The results of the present study indicated that the activity levels of AST and the concentrations of urea and creatinine remained unaffected following combined treatment with all examined extracts (*p* > 0.05). However, the treatment with *Lycopodium clavatum* L., *Lycopodium annotinum* L. and *Huperzia selago* L. resulted in a decrease in GGT activity levels in the rat serum compared with control (F[2.45] = 7.571, *p* < 0.01, *p* < 0.01, *p* < 0.01). Likewise, the treatment with *Lycopodium annotinum* L. and *Huperzia selago* L. resulted in a decrease in ALT activity levels in the rat serum compared with control (F[2.45] = 2.501, *p* < 0.05, *p* < 0.05).

#### 3.3.2. ACHE, BCHE Activity; IL-1β and IL-6 Concentrations

The effects of plant extracts treatment on the AChE activity in the serum and prefrontal cortex of rats are shown in Figure 4a and Figure 5a. A statistically significant decrease in AChE activity as compared to the control group was observed in the blood serum of rats receiving *Lycopodium annotinum* L. (F[2.64] = 2.147, *p* < 0.05) and in the prefrontal cortex of rats receiving *Angelica dahurica* (F[2.64] = 2.977, *p* < 0.001). *Angelica dahurica* and *Lycopodium annotinum* L. extract also caused a statistically significant reduction in BChE activity in the serum of rats as compared to the control group (F[2.64] = 6.317, *p* < 0.05, *p* < 0.05; Figure 4b). However, no statistically significant changes in BChE activity were observed in the prefrontal cortex of rats compared to controls (Figure 5b). The administration of examined extracts did not influence the concentrations of IL-1β nor IL-6 in the serum of rats (Figure 4c,d), but one-way ANOVA analysis showed a statistically significant decrease in the concentration of IL-1β in the brain of all groups of rats receiving the test extracts as compared to the control (F[2.64] = 8.494, *p* < 0.05, *p* < 0.05, *p* < 0.01, *p* < 0.01, *p* < 0.01, respectively; Figure 5c). A statistically significant reduction in the level of IL-6 as compared to the control was also noted in the prefrontal cortex of animals receiving *Angelica dahurica* (F[2.64] = 3.455, *p* < 0.05; Figure 5d).

## 4. Discussion

As mentioned in the introduction, no studies confirm the safety and pharmacological activity of *Narcissus triandrus* L., *Lycopodium clavatum* L., *Lycopodium annotinum* L., *Huperzia selago* L. and *Angelica dahurica* (sp.) extracts. Studies that inspired the search for potent AChE inhibitors showed that galantamine (an alkaloid found in the bulbs and flowers of some members of the *Amaryllidaceae* family, such as *Narcissus)* prevents cognitive deficits, including learning and spatial memory deficits, as well as passive avoidance memory acquisition in rodents [44]. This is confirmed by clinical studies, according to which long-term treatment with galantamine improves cognitive deficits in patients with Alzheimer’s disease [45]. Huperzine A (HupA), a natural compound contained in the Chinese moss *Huperzia serrata* (Thunb. Ex Murray) Trevis (family *Lycopodiaceae*), has an AChE inhibitory effect [46,47,48]. Similarly, in vivo studies showed that *Lycopodium selago* extract, administered to zebrafish (*Danio rerio*) with cognitive impairment caused by scopolamine, alleviates cognitive deficits, inter alia, by regulating ACHE activity [46]. Another study confirmed that treatment with HupA improved cognition in mice with diet-induced obesity [49]. Other studies showed that angelica polysaccharide ameliorates memory impairment in animal models of cognitive disorders, although the exact mechanisms have not yet been understood [50,51,52,53].

Phytochemical analysis of species of the *Lycopodiaceae* family is still a subject of much research. The group of compounds with the strongest potential are alkaloids. These compounds are characterized by a diversity of structure and biological activity. The typical *Lycopodiaceae* alkaloids are composed of two piperidine rings belonging to quinolizine, or pyridine and α-pyridone type alkaloids. Many of the studies on the *Lycopodiaceae* alkaloids were carried out in the second half of the 20th century. They led to the isolation of more than 200 alkaloids, which were then divided by Canadian scientists in the 1990s into four major classes based on similarities in structure and pharmacological activity, including lycopodine, lycodine, fawcettimine and miscellaneous [34]. Alkaloids belonging to the lycopodine type with a single nitrogen atom in their structure constitute the largest group with more than 100 compounds. However, most of the isolated alkaloids with outstanding AChE inhibitory activity belong to the class of lycodine, which have two nitrogen atoms in the structure, such as the well-known huperzine A, which has great potential in the treatment of Alzheimer’s disease. Phytochemical analysis of the methanolic extracts and alkaloid fractions obtained during the study confirmed the presence of both known and new potent AChE inhibitors [34,37]. Apart from alkaloids, the presence of flavonoid compounds, luteolin and chrysoeriol; phenolic acids, ferulic, chlorogenic or triterpenes; and mineral salts is also described in the literature [47]. *Narcissus species* is also a source of bioactive compounds belonging to the isoquinoline alkaloid group. Many years of research have allowed the isolation of more than 100 *Amaryllidaceae* alkaloids from the *Narcissus genus*, which are divided into different groups depending on the structural composition of these compounds [54,55]. The classification, according to Jin and Yao, 2019, is at present the most current proposed division of the structure of *Amaryllidaceae* alkaloids, presenting 25 types of alkaloids, including five main groups: lycorine, crinine, cherylline, buflavine and galanthamine [56]. Phytochemical analysis conducted allowed confirming the presence of some of these compounds in particular fractions obtained from the tested methanolic extract of *Narcissus hawera* [38]. Many alkaloids were identified, including sanguinine, which proved to be an up to 10 times more potent inhibitor than the known galantamine in the course of in vitro tests [32]. Non-alkaloid components, including flavonoids, chalcones, lignans or triterpenes were also reported in the literature [57]. Radix *Angelicae dahuricae*, the dry root of *Angelica dahurica* (Boiss.), is listed in the Chinese Pharmacopoeia. It is a rich source of coumarins and furanocoumarins, among them coumarin, scopoletin, psoralen, xanthotoxin, bergapten, isoimperatorin imperatorin byakangelicol, oxypeucedanin and phellopterin [58,59]. These compounds were reported to exhibit pharmacological effects such as inhibition of acetylcholinesterase [60] and inhibitory effects on GABA transaminase responsible for degradation of this neurotransmitter [61]. Imperatorin was chosen as the only marker for controlling the quality of *Angelicae dahuricae* radix in the 2010 edition of the Chinese Pharmacopoeia [62], which indicates that the content of imperatorin in *Angelicae dahuricae* radix should not be less than 0.08%, otherwise it should not be used as a medicinal material for clinical application [63].

The assessment of safety is of key importance in considering further research on the efficacy and potential therapeutic application of new compounds. Since the safety profile of the extracts used in the study has never been determined before, we decided to determine the parameters proving the function of the liver and kidneys. In addition, the animals were monitored daily by a veterinarian, and body weight was measured every 7 days. For the results to be fully reliable, the tested extracts were administered to healthy rats without causing cognitive impairment. The observation of the animals and the results of their body weight measurements indicate the safety of the tested substances. There were no statistically significant changes in the body weight of rats treated with plant extracts for 28 days compared to the control group. Similarly, the appearance and behavior of the animals in the test groups did not differ from the control group. All animals survived the experiment in very good condition.

The results of this study showed that the tested extracts did not significantly affect the levels of AST activity compared to the control group. In the groups of animals that received *Lycopodium clavatum* L., *Lycopodium annotinum* L. and *Huperzia selago* L., a decrease in GGT activity was noted as compared to the control group, while in the groups receiving *Lycopodium annotinum* L., *Huperzia selago* L. decreased ALT activity was observed. GGT is a cell surface enzyme. The highest levels are in the kidneys, intestines and liver. The liver produces most of the GGT in the blood [64]. The primary function of GGT is to break down and recycle glutathione, the most important antioxidant in the human body, which increases the number of available amino acids (especially cysteine) that are used for glutathione inside the cell [65]. GGT may increase oxidative stress through the breakdown of glutathione (and the production of cysteinylglycine), which leads to tissue, cell and DNA damage. Increased levels of GGT are evidence of impaired liver function and increase the risk of many diseases, incl. cardiovascular diseases, kidney diseases, cancer and metabolic syndrome [66,67]. GGT may also be elevated in patients with Alzheimer’s disease. A clinical trial involving 2.4 thousand men revealed that an increased GGT level might be associated with an increased risk of dementia [68]. However, research does not indicate that low GGT levels can be dangerous to the body. Reduced levels of this enzyme activity may be hereditary or may affect women in the second and third trimesters of pregnancy [69]. Thus, the changes observed in our study in GGT activity do not indicate a risk of impaired liver function; on the contrary, in the case of the three tested extracts, they may prove beneficial, given the unfavorably increased levels of GGT in demented patients. This, however, requires further research. ALT is the most widely used clinical biomarker of liver function. This enzyme is involved in the transamination of alanine and is present in the liver in much higher concentrations than in other organs. Leakage of ALT from hepatocytes into the blood follows damage to hepatic cells [70]. Although elevated ALT levels may strongly suggest liver damage, ALT is not liver-specific, and extrahepatic sources of circulating ALT also include areas of skeletal muscle or myocardial damage [71]. As in the case of GGT, low ALT levels do not indicate any organ dysfunction and are not harmful to the body. For a complete and accurate interpretation of the obtained test results, especially changes in the decrease in the activity/concentration of the determined parameters, further studies are needed, also taking into account the findings before starting the experiment.

The results of this study also showed that 28-day administration of the test extracts to rats did not cause significant changes in markers of renal function—in urea and creatinine concentrations compared to the control group. Therefore, one can conclude about the safety of the tested plant extracts in the context of internal organs—liver and kidneys.

In this experiment, behavioral tasks were performed to evaluate the effects of the test extracts on the central nervous system of rats. Our study showed that 28 days of administration of the test extracts to rats did not affect their locomotor activity in the locomotor activity test: they neither stimulated nor weakened motor activity. The willingness of rodents exposed to the test extracts to discover new environments was measured in the Y-maze spontaneous alternation behavioral test. Many parts of the brain—including the prefrontal cortex—are involved in this task. Animals have a natural tendency to explore. This tendency is exploited, inter alia, in the Y-maze test. When rats are allowed to freely explore the Y-maze, rats tend to explore each arm rather than returning to just the same arm. In our study, based on this test, rodent motor activity was assessed by measuring the total number of arm entries and the evaluation of fresh spatial memory by calculating % logical alternations. The results of our research revealed that in the experimental groups, spontaneous alternation was in the chance level (50%), which is indicative of functional working memory. This memory can be impaired in a wide variety of disorders, such as Alzheimer’s disease and dementia. In our study, the extracts were administered to animals without first inducing memory deficits, which explains the results obtained. Our results also revealed that the spontaneous alternation percentage of arm entries significantly increased in the *Huperzia selago* L. group, which may indicate an improvement in fresh spatial memory, even in healthy rats. Moreover, we observed that the total number of arm entries as compared to the control was significantly higher in the group of rats receiving *Narcissus triandrus* L., which in turn may indicate an increase in locomotor activity in this group. The remaining extracts showed no effect on the cognitive functions of rats: they did not damage them but also did not cause cognitive deficits. Taking into account the fact that the tested extracts were administered to healthy rats without induced cognitive impairment (examination of the extracts in the animal AD model is planned in the next stage), the obtained results should be interpreted as a positive effect on the safety assessment. Similarly, the analysis of the results of the passive avoidance test showed that none of the examined extracts caused disturbances in cognitive processes and short-term memory in rats. On the contrary, a positive effect of the extracts of *Narcissus triandrus* L., *Lycopodium clavatum* L., *Huperzia selago* L. and *Angelica dahurica* on learning and memory processes were observed. Twenty-eight days of administration of these extracts to animals significantly increased STL, which may indicate a significant effect of these substances on improving memory.

As is known, AD is characterized by a decrease in the concentration of ACh in cholinergic neurons. Therefore, it is very important to increase the level of ACh, which can be obtained by inhibiting the ACh-hydrolyzing enzyme, which in turn leads to an increase in its concentration in the synapses of the central nervous system. The intensification of neurotransmission in the cholinergic system inhibits the symptoms of the disease. We determined the activity of cholinesterases in the serum and prefrontal cortex of rats treated with the extracts. The results revealed a significant reduction in AChE activity in the serum of rats receiving *Lycopodium annotinum* L. and in the prefrontal cortex of rats receiving *Angelica dahurica*. Moreover, *Angelica dahurica* extract also caused a significant reduction in BChE activity in the serum of rats, while the BChE activity in the prefrontal cortex of rats remained unchanged. AChE is a specific esterase belonging to the carboxylesterase family of enzymes. It mainly hydrolyses acetylcholine. AChE is found in high concentrations, mainly in red blood cells and also in the brain at neuromuscular junctions and cholinergic synapses of the brain. BChE, also known as ‘pseudo’ cholinesterase, plasma, non-specific or type II cholinesterase, is a non-specific cholinesterase enzyme that hydrolyses various types of choline esters [72]. This enzyme is present throughout the body, including blood serum and the central nervous system. In the brain, BChE is mainly associated with endothelial cells and glial cells [73,74,75]. Under physiological conditions, BChE plays a supporting role in the brain and is responsible for approximately 10% of cholinesterase activity [73]. Like AChE, the enzyme BChE is involved in the hydrolysis of the neurotransmitter ACh and was therefore proposed as a viable therapeutic target in AD, a disorder characterized by a cholinergic deficit. AChE and BChE show different kinetic responses to ACh concentrations. At low concentrations of ACh, AChE is more active than BChE, and at higher concentrations of ACh, the activity of BChE in ACh hydrolysis is significantly increased [76]. In our study, we were going to check whether the tested extracts affected the activity of cholinesterases in the rat brain; for comparison, we also performed blood tests. The results that show the positive potential of *Angelica dahurica* in this regard are promising. Administration of this extract to animals for 28 days resulted in a significant reduction in AChE activity in the brain. The lack of influence on the activity of BChE may result from its low activity in the brain in its physiological state because under normal conditions, BChE plays only a supporting role in the brain. However, it should be noted that *Angelica dahurica* lowered the activity of BChE in the blood serum of the animals. BChE is a plasma cholinesterase, which may explain the obtained results. To precisely check the influence of the tested extracts on the activity of BChE in the brain, it is necessary to conduct studies on animals with induced dementia. BChE plays a key role in the nervous system, especially in the co-regulation of ACh levels in the brain in neurodegenerative disorders such as AD [73,77]. In AD, AChE levels decrease by as much as 85% in some areas of the brain, while BChE levels increase with disease progression [78,79]. Perhaps, under such conditions, the influence of the tested extracts, and especially *Angelica dahurica*, on the activity of enzymes would be significant, which may suggest the obtained result of lowering the activity of BChE in the blood.

A growing body of evidence indicates that AD neurotoxicity is mediated by inflammatory processes in the central nervous system. These processes include activation of microglia by amyloid-β, leading to the release of pro-inflammatory cytokines, including IL-1β and IL-6 [80]. The neurotoxic processes associated with the release of pro-inflammatory cytokines may include direct neuronal death by enhancing apoptosis and decreased synaptic function by inhibiting hippocampal neurogenesis. There is increasing evidence that major pro-inflammatory cytokines such as IL-1-β and IL-6 can lead to death and neuronal dysfunction in the brain. Cytokines can also affect the synaptic function and plasticity of the brain [81]. Central nervous system inflammation may predate the development of senile plaques and neurofibrillary tangles in dementia [80]. The cholinergic anti-inflammatory pathway can be impaired by AChE and BChE, which hydrolyze and inactivate AChE, thereby antagonizing vagal cholinergic signaling at the macrophage level with the release of pro-inflammatory cytokines [82]. Increased activity of AChE and BChE leads to an increase in hydrolytic degradation and a decrease in the concentration of the neurotransmitter, which in turn may cause systemic inflammation [83]. ACH is a neurotransmitter and regulates the level and activity of serotonin, dopamine and other neuropeptides, thus modulating not only neurotransmission but also the immune response. Hence, both AChE and BChE can increase inflammation by inactivating acetylcholine. However, clinical trials of nonsteroidal anti-inflammatory drugs (NSAIDs) have not provided sufficient benefit. This is likely since the relationship between innate immunity and AD pathogenesis is complex, and the immune response can be both beneficial and harmful, depending on the context [9,20,84]. Therefore, it seems reasonable to look for drugs that both inhibit cholinesterase activities and simultaneously reduce inflammation in the brain by lowering the level of pro-inflammatory cytokines. The results of our research showed a statistically significant decrease in the concentration of IL-β in the brain of all groups of rats receiving the test extracts and also a decrease in the concentration of IL-6 in the brain of rats receiving *Angelica dahurica*. However, we did not observe any statistically significant changes in the level of these cytokines in the blood serum of rats exposed to the tested extracts. When taking into account the above considerations, a significant decrease in the level of pro-inflammatory IL-β in the brain of animals by all tested extracts, as well as a decrease in the level of pro-inflammatory IL-6 by *Angelica dahurica*, with a simultaneous, also significant, reduction in AChE activity, indicate a beneficial potential in this aspect and is the basis for further research in the AD model. The observed lack of changes in the level of IL-6 in the brain and the lack of changes in the concentration of both cytokines in the blood may be due to the fact that the experiment was carried out on healthy animals that did not induce inflammation and did not increase pro-inflammatory cytokines. However, this may also be due to the selectivity in the action of the test substances; it requires confirmation/verification in subsequent tests. However, the lack of influence of the tested extracts on the concentration of cytokines in the blood, with the simultaneous statically significant fluid on their concentration in the brain, seems to be surprising. Perhaps the explanation of the results obtained is also related to the mechanisms of influence on neurotransmitters and inflammatory processes. It is known that ACh is anti-inflammatory, and the acetylcholine receptor modulates the interaction between the immune system and the nervous system [73]. Thus, lowering the activity of cholinesterases increases the concentration of ACh, which may, in turn, affect the production of pro-inflammatory markers. Moreover, in vitro studies have shown that selective inhibition of BChE reduces the production of pro-inflammatory cytokines such as iIL-1β in peripheral blood mononuclear cell cultures [85]. The results of our research, which show that the studied extracts, and in particular *Angelica dahurica*, can modulate the levels of pro-inflammatory cytokines in the brain (also in the physiological state), and which may be of particular importance in AD, are very promising and may constitute the basis for further research in to demonstrate their mechanisms of action in the dementia process.

## 5. Conclusions

The observations of animals, the results of behavioral tests, measurements of their body weight and the analysis of parameters indicating the function of the liver and kidneys indicate the safety of the tested substances. The conducted behavioral tests showed that the tested extracts did not significantly affect the motor coordination of animals, but they improved their memory. The results of biochemical studies showed that *Angelica dahurica* extract has the most promising potential for future AD therapeutic strategies by influencing ACh levels and inflammatory processes in the brain.

## Figures and Tables

**Figure 1 jcm-11-04291-f001:**
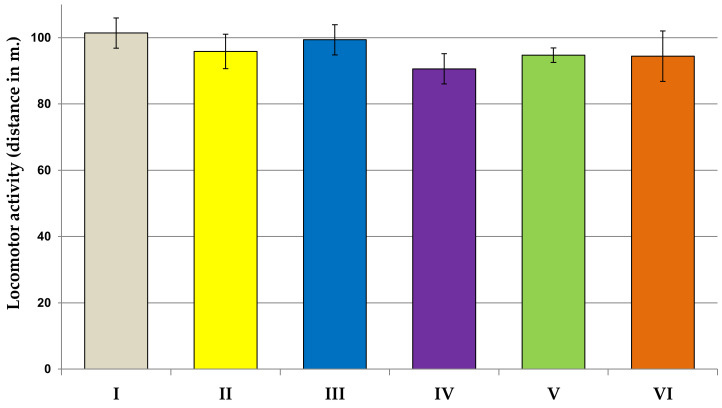
The influence of extracts on locomotor activity. All values are reported as mean ± SEM (*n* = 7). I—*aqua pro injectione*; II—*Narcissus triandrus* L. 20 mg/kg; III—*Lycopodium clavatum* L. 10 mg/kg; IV—*Lycopodium annotinum* L. 10 mg/kg; V—*Huperzia selago* L. 10 mg/kg; VI—*Angelica dahurica* (sp.) 200 mg/kg.

**Figure 2 jcm-11-04291-f002:**
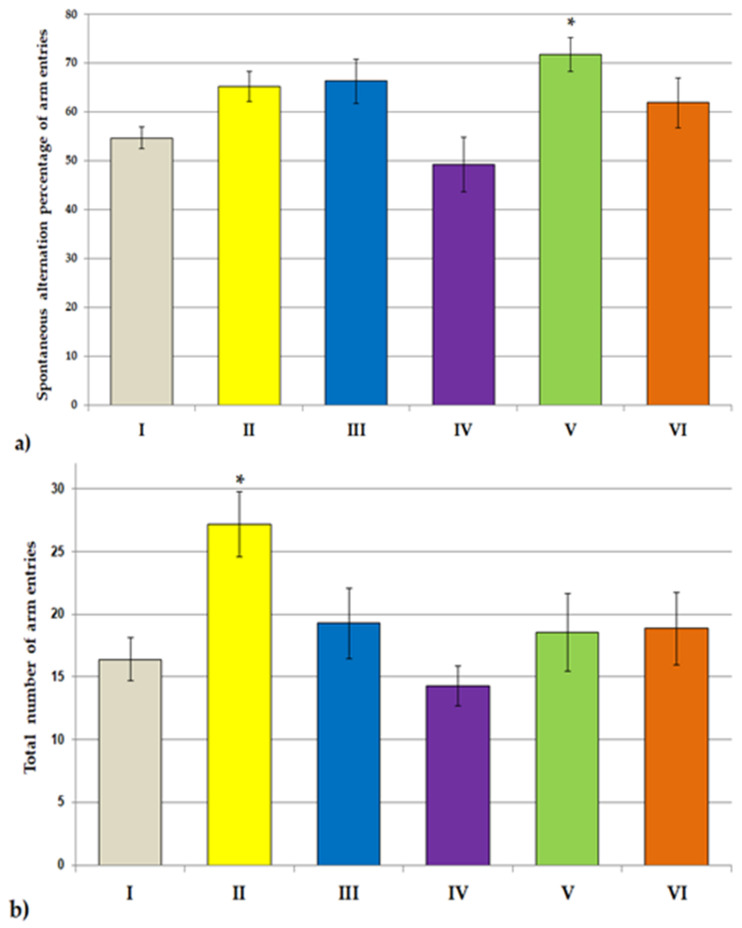
The influence of extracts on the spontaneous alternation behavior and spatial memory in Y-maze test. The percentage of spontaneous transformation (**a**) and the total number of entries into the arm (**b**) were determined. All values are reported as mean ± SEM (*n* = 7). Significance: * *p* < 0.05, vs. control group (one-way ANOVA followed by Tukey’s *post hoc* test). I—*aqua pro injectione*; II—*Narcissus triandrus* L. 20 mg/kg; III—*Lycopodium clavatum* L. 10 mg/kg; IV—*Lycopodium annotinum* L. 10 mg/kg; V—*Huperzia selago* L. 10 mg/kg; VI—*Angelica dahurica* (sp.) 200 mg/kg.

**Figure 3 jcm-11-04291-f003:**
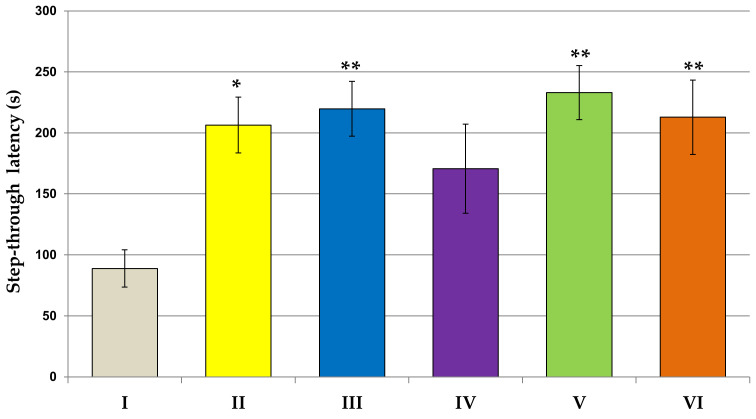
The influence of extracts on the learning and memory performance in passive avoidance test. All values are reported as mean ± SEM (*n* = 7). Significance: * *p* < 0.05, ** *p* < 0.01 vs. control group (one-way ANOVA followed by Tukey’s *post hoc* test). I—*aqua pro injectione*; II—*Narcissus triandrus* L. 20 mg/kg; III—*Lycopodium clavatum* L. 10 mg/kg; IV—*Lycopodium annotinum* L. 10 mg/kg; V—*Huperzia selago* L. 10 mg/kg; VI—*Angelica dahurica* (sp.) 200 mg/kg.

**Figure 4 jcm-11-04291-f004:**
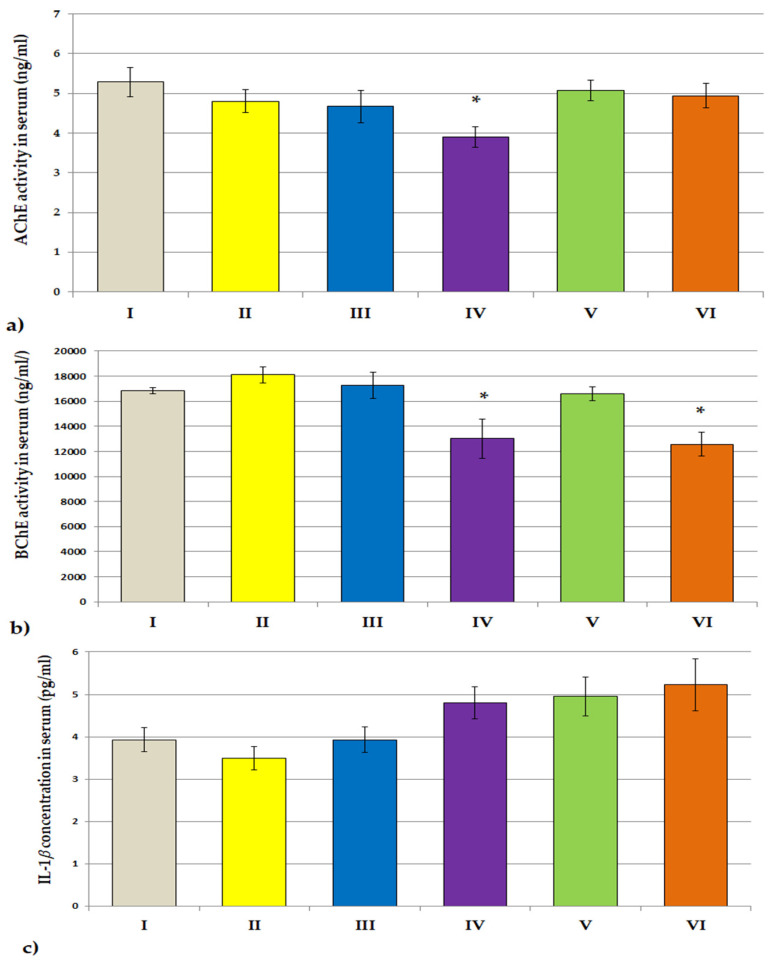
The influence of extracts on the ACHE (**a**) activity, BCHE activity (**b**), IL-1β concentrations (**c**) and IL-6 concentrations (**d**) in the serum of rats. All values are reported as mean ± SEM (*n* = 7). Significance: * *p* <0.05 vs. control group (one-way ANOVA followed by Tukey’s *post hoc* test). I—*aqua pro injectione*; II—*Narcissus triandrus* L. 20 mg/kg; III—*Lycopodium clavatum* L. 10 mg/kg; IV—*Lycopodium annotinum* L. 10 mg/kg; V—*Huperzia selago* L. 10 mg/kg; VI—*Angelica dahurica* (sp.) 200 mg/kg.

**Figure 5 jcm-11-04291-f005:**
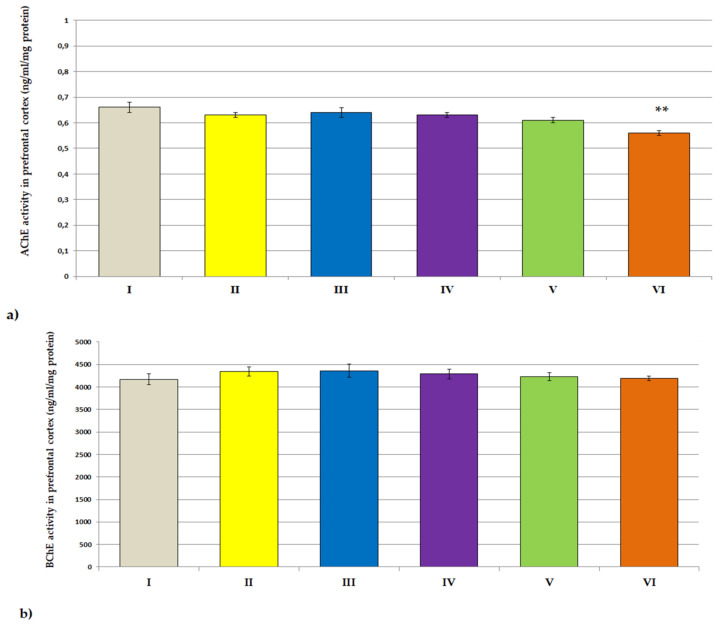
The influence of extracts on the ACHE (**a**) activity, BCHE activity (**b**), IL-1β concentrations (**c**) and IL-6 concentrations (**d**) in the prefrontal cortex of rats. All values are reported as mean ± SEM (*n* = 7). Significance: * *p* < 0.05, ** *p* < 0.01 vs. control group (one-way ANOVA followed by Tukey’s post hoc test). I—*aqua pro injectione*; II—*Narcissus triandrus* L. 20 mg/kg; III—*Lycopodium clavatum* L. 10 mg/kg; IV—*Lycopodium annotinum* L. 10 mg/kg; V—*Huperzia selago* L. 10 mg/kg; VI—*Angelica dahurica* (sp.) 200 mg/kg.

**Table 1 jcm-11-04291-t001:** The average weight of rats [g] measured every 7 days during the experiment. Data are presented as the means ± SEM (*n* = 7).

Treatment	Day 1	Day 7	Day 14	Day 21	Day 28
**Control group**	257.5 ± 5.26	281.1 ± 6.5	299.4 ± 6.9	316.2 ± 5.8	333.4 ± 5.9
***Narcissus triandrus* L.**	283.7 ± 13.5	291.8 ± 9.3	303.1 ± 24.4	333.0 ± 10.8	346.4 ± 11
***Lycopodium clavatum* L.**	265.5 ± 10.6	283.7 ± 11.6	296.1 ± 12.9	306.57 ± 12.5	319.2 ± 12
***Lycopodium annotinum* L.**	257.1 ± 4.5	279.7 ± 4.4	291.4 ± 4.4	302.8 ± 4.4	302.7 ± 13
***Huperzia selago* L.**	276.0 ± 6.2	295.0 ± 7.6	313.0 ± 9.4	325.2 ± 9.9	334.1 ± 9.6
***Angelica dahurica* (sp.)**	255.0 ± 8.1	277.7 ± 10.9	294.1 ± 13.1	305.7 ± 13	321 ± 12.5

**Table 2 jcm-11-04291-t002:** The influence of extracts on the AST, ALT and GGT activity and urea and creatinine concentrations. All values are reported as mean ± SEM (*n* = 7). Significance: * *p* < 0.05, ** *p* < 0.01 vs. control group (one-way ANOVA followed by Tukey’s post hoc test).

Treatment	AST [U/L]	ALT [U/L]	GGT [U/L]	UREA [mg/dL]	Creatinine [mg/dL]
**Control group**	77.57 ± 2.30	29.43 ± 2.5	5.39 ± 0.95	37.93 ± 8.8	0.32 ± 0.05
***Narcissus triandrus* L.**	63.52 ± 5.98	24.51 ± 1.5	3.90 ± 0.29	33.80 ± 4.3	0.34 ± 0.037
***Lycopodium clavatum* L.**	62.2 ± 1.89	25.6 ± 0.98	2.77 ± 0.23 **	44.7 ± 2.7	0.30 ± 0.02
** *Lycopodium* ** ***annotinum* L.**	65.03 ± 8.5	21.75 ± 0.97 *	1.88 ± 0.37 **	36.96 ± 1.6	0.28 ± 0.01
***Huperzia selago* L.**	60.45 ± 3.9	24.46 ± 1.1 *	2.51 ± 0.36 **	41.09 ± 4.2	0.29 ± 0.01
***Angelica dahurica* (sp.)**	61.11 ± 3.47	26.33 ± 3.05	3.96 ± 0.22	26.75 ± 2.7	0.28 ± 0.02

## Data Availability

Not applicable.

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
