# Peer review of "Amaryllidaceae, Lycopodiaceae Alkaloids and Coumarins—A Comparative Assessment of Safety and Pharmacological Activity"

_jcm, 2022, doi:10.3390/jcm11154291_

Round 1

Reviewer 1 Report

The manuscript jcm-1803894 investigates the effects of natural compounds on various parameters of rats such as behavioural tests, activity of acetylcholinesterase (AChE) and butyrylcholinesterase (BChE), inflammatory markers (IL-1beta and IL-6), liver and kidney function. According to the title, the tested compounds could be considered as therapeutic agents for Alzheimer’s disease, but actually the animal model used here is not reproducing the disease. Rather, healthy animals were used. Therefore, the title does not truly reflect the goal of the work, which might as well demonstrate the safety but not the effectiveness of the tested natural compounds on the disease. Additional comments are as follows.

“Phytochemical investigation of the plant extracts” also contains unnecessary background information which is not pertinent to this paragraph. More importantly, the results of this analysis are not presented in the manuscript, being the chemical characterization completely lacking.

The way figures are arranged should be reconsidered, trying to group them as appropriate.

Author Response

Response to Reviewer 1 Comments

Point 1: The manuscript jcm-1803894 investigates the effects of natural compounds on various parameters of rats such as behavioural tests, activity of acetylcholinesterase (AChE) and butyrylcholinesterase (BChE), inflammatory markers (IL-1beta and IL-6), liver and kidney function. According to the title, the tested compounds could be considered as therapeutic agents for Alzheimer’s disease, but actually the animal model used here is not reproducing the disease. Rather, healthy animals were used. Therefore, the title does not truly reflect the goal of the work, which might as well demonstrate the safety but not the effectiveness of the tested natural compounds on the disease. Additional comments are as follows.

Response 1: We would like to thank the Reviewer for comments and suggestion, which helped us improve the quality of our manuscript. The manuscript contains the results of the first research stage, i.e. determination of safety and pharmacological activity of substances with properties that may allow them to be used in the treatment of Alzheimer's disease in the future. This requires testing on healthy animals. We are currently conducting research of isolated fractions from the most promising extracts in a rat model of Alzheimer's disease. We agree with the Reviewer that the title of the work does not fully reflect the purpose of the work and therefore it has been changed.

Point 2: “Phytochemical investigation of the plant extracts” also contains unnecessary background information which is not pertinent to this paragraph. More importantly, the results of this analysis are not presented in the manuscript, being the chemical characterization completely lacking.

Response 2: After taking into account the Reviewer's suggestions, the chapter has been transferred into the Discussion part. The extracts are chemically defined according to our previous published papers. Therefore, no additional data are not necessary to be presented.

Point 3: The way figures are arranged should be reconsidered, trying to group them as appropriate.

Response 3: The scheme of presenting the results in the figures and the table is, in our opinion, clearly presented: figures 1, 2a and b and 3 refer to behavioral tests, the evaluation of parameters indicating the function of the liver and kidneys is presented in the table 2, while the results of biochemical tests are presented in figures 4-7. Figures 4a-7a show the results of the serum tests and figures 4b-7b the results of the prefrontal cortex tests. After taking into account the comments of the Reviewers, the presentation of the figures was changed to make it more legible. The signatures and legends of the figures were also changed. However, we are open to suggestions or comments from the Reviewer and the Editor as to the method of presenting research results and ready to make suitable adjustments.

Reviewer 2 Report

Dear authors,

I enjoyed reading the manuscript entitled "Amaryllidaceae, Lycopodiaceae alkaloids and coumarins as new therapeutic strategies for Alzheimer's disease-comparative assessment of safety and activity". The paper focuses on the active evaluation of the safety and activity of compounds of plant origin, rich in alkaloids and coumarin derivatives, as well as their beneficial effects in the therapy of Alzheimer's Dementia.

The manuscript is valuable because according to research, it would be the first study to evaluate the safety and activity of compounds, using state-of-the-art techniques. The many hypotheses underlying Alzheimer's Dementia are a challenge for discovering new therapeutic strategies. As you mentioned the cholinergic hypothesis plays an important role in A.D. and plant compounds from the Amaryllidaceae and Lycopodiaceae families could be therapeutic targets. The paper is presented in a clear and concise manner, respecting the requirements of the journal. The bibliographic references are in large numbers for an original work, which shows the in-depth study of the literature on this subject. In the chapter Material and Method, all the work stages were presented in detail. The results obtained were presented in tables and figures, thus allowing the manuscript to be read easily. Although they were not mandatory, I appreciate the fact that you also gave us some pertinent conclusions.

However, I have a number of comments:

1. In point 2.5. you mentioned that you used Tween 80. It was most likely used to promote the solubilization of plant extracts when preparing solutions for oral administration. What is the maximum allowable limit for Tween 80 when preparing such solutions? What volume can be added?

2. What was the volume of solution administered daily to rats in the case of treatments?

3. Table 2 presents a series of biochemical markers. In most cases, the values obtained are higher in the case of the control group than in the case of the treated lots. How do you explain this? I wonder if it would have been more appropriate to make an assessment of these parameters even before starting the administrations with the respective extracts.

4. A weak point as you mentioned in the paper is the absence of the dementia model. The correct evaluation of all parameters is done by comparing the results between the control group and those treated with plant extracts but also comparing with those on which the model was induced. Is comparison only with the control group sufficient?

5. You administered the compounds for a period of 28 days. You only rated the short-term memory. Such administrations would also allow you to analyze long-term memory. Have you somehow taken this into account for a future study?

6. Some coumarin derivatives are known for their anticoagulant effects. In the present case, did the administration for 28 days have any impact on the bleeding and coagulation time in the experimental animals? Were any bleeding or other phenomena observed?

7. For all 7 figures, I recommend that the writing under each batch be more legible.

8. English, although quite good, could be improved.

9. On line 184 there is a parenthesis marked in yellow, immediately after reference 46.

Author Response

Response to Reviewer 2 Comments

Dear authors,

I enjoyed reading the manuscript entitled "Amaryllidaceae, Lycopodiaceae alkaloids and coumarins as new therapeutic strategies for Alzheimer's disease-comparative assessment of safety and activity". The paper focuses on the active evaluation of the safety and activity of compounds of plant origin, rich in alkaloids and coumarin derivatives, as well as their beneficial effects in the therapy of Alzheimer's Dementia.

The manuscript is valuable because according to research, it would be the first study to evaluate the safety and activity of compounds, using state-of-the-art techniques. The many hypotheses underlying Alzheimer's Dementia are a challenge for discovering new therapeutic strategies. As you mentioned the cholinergic hypothesis plays an important role in A.D. and plant compounds from the Amaryllidaceae and Lycopodiaceae families could be therapeutic targets. The paper is presented in a clear and concise manner, respecting the requirements of the journal. The bibliographic references are in large numbers for an original work, which shows the in-depth study of the literature on this subject. In the chapter Material and Method, all the work stages were presented in detail. The results obtained were presented in tables and figures, thus allowing the manuscript to be read easily. Although they were not mandatory, I appreciate the fact that you also gave us some pertinent conclusions.

However, I have a number of comments:

Point 1: In point 2.5. you mentioned that you used Tween 80. It was most likely used to promote the solubilization of plant extracts when preparing solutions for oral administration. What is the maximum allowable limit for Tween 80 when preparing such solutions? What volume can be added?

Point 2: What was the volume of solution administered daily to rats in the case of treatments?

Response 1 and 2: We would like to thank the Reviewer for comments and suggestion, which helped us improve the quality of our manuscript. Tween 80 (polysorbate 80, polyoxyethylene sorbitan monooleate) is a non-ionic surfactant that is widely used as an emulsifier in cosmetics, pharmaceuticals and food products. It is approved by the US Food and Drug Administration for use in up to 1% in selected foods. Tween 80 has been used to promote the solubilisation of plant extracts in the preparation of solutions for oral administration. We added 1-2 drops of 1% Tween 80 and the solutions of extracts prepared in this way we administered in the amount of 1 ml per 100 g of the rat's body weight. In the manuscript, we added the sentence: ”Plant extracts were suspended in 1% Tween 80 in saline and the solutions prepared in this way and then were administered per os  in a volume of 1 mL/100 g of the body weight.” We based both the selection of Tween 80 concentration and the volume of the administered solutions on appropriate guidelines and data contained in the literature. We apologize for the mistake, the error has been corrected in the manuscript.

Paul Désiré DD, Yolande Sandrine MN, Danielle Claude B, Mireille K, Oumarou Bibi-Farouck A, Théophile D, Pierre K. In vivo estrogenic-like activities of Gouania longipetala Hemsl. (Rhamnaceae) bark extracts in a post-menopause-like model of ovariectomized Wistar rats. J Ethnopharmacol. 2015 Jun 20;168:122-8. doi: 10.1016/j.jep.2015.03.049. Epub 2015 Apr 4. PMID: 25849733.

https://iqconsortium.org/images/LG-3Rs/IQ-CRO_Recommended_Dose_Volumes_for_Common_Laboratory_Animals_June_2016_%282%29.pdf

Point 3: Table 2 presents a series of biochemical markers. In most cases, the values obtained are higher in the case of the control group than in the case of the treated lots. How do you explain this? I wonder if it would have been more appropriate to make an assessment of these parameters even before starting the administrations with the respective extracts.

Response 3: As all the animals used for the study were healthy and came from a certified breeder, we assumed that there is no need to perform the biochemical determinations of liver and kidney function before starting the experiment. However, as the reviewer noted, some of the results were surprising to us as well. In the groups of animals receiving Lycopodium clavatum L., Lycopodium annotinum L. and Huperzia selago L. a decrease in GGT activity was noted compared to the control group, while in the groups receiving Lycopodium annotinum L., Huperzia selago L. decreased ALT activity was found. It is known that increased GGT levels are evidence of impaired liver function and increase the risk of many diseases, including Cardiovascular disease and kidney disease may also be increased in Alzheimer's patients. However, studies do not indicate that low GGT levels could be dangerous to the body. Therefore we found that the changes in GGT activity observed in our study do not indicate a risk of impaired liver function and, on the contrary, in the case of the three tested extracts, they may be beneficial considering unfavorably elevated GGT levels in dementia of patients. Yet, this requires further research. As with GGT, a low ALT level is also not indicative of organ dysfunction and is not harmful to the body. Notwithstanding, we agree with the Reviewer that these results are surprising. For their full interpretation, further research is required, taking into account the determinations before the start of the experiment. This information has been included in the discussion chapter.

Point 4: A weak point as you mentioned in the paper is the absence of the dementia model. The correct evaluation of all parameters is done by comparing the results between the control group and those treated with plant extracts but also comparing with those on which the model was induced. Is comparison only with the control group sufficient?

Response 4: We agree with the Reviewer that in order to correctly assess all parameters, the results should be compared between the control group and the group treated with plant extracts and compared with those on which the model was induced. The manuscript submitted for review contains the results of the first stage of research - determination of the safety and pharmacological activity of substances with properties that may allow them to be used in the treatment of Alzheimer's disease in the future. This requires testing on healthy animals. We are currently conducting research of isolated fractions from the most promising extracts in a rat model of Alzheimer's disease. Only a comparison of the safety and pharmacological activity of both studies will provide reliable information. However, to be able to move to the second stage of the research, we needed preliminary results that allowed us to select the most appropriate fractions in terms of safety and bioactivity.

Point 5: You administered the compounds for a period of 28 days. You only rated the short-term memory. Such administrations would also allow you to analyze long-term memory. Have you somehow taken this into account for a future study?

Response 5: We would like to thank the Reviewer for this remark. In future studies, we included the assessment of long-term memory in behavioral tests: in the Passive Avoidance Test at 24, 48, and 72 hours, in the Novel Object recognition (NOR) task, which is used to evaluate cognition, particularly recognition memory, in rodent models of CNS disorders and in the Morris Water Maze (MWM), which is designed to test spatial memory and long term memory by observing and recording escape latency, distance moved and velocity during the time spend in the water tank.

Point 6: Some coumarin derivatives are known for their anticoagulant effects. In the present case, did the administration for 28 days have any impact on the bleeding and coagulation time in the experimental animals? Were any bleeding or other phenomena observed?

Response 6: During our experiment, neither we nor the attending veterinarian observed any signs of bleeding or any disturbing phenomena in the experimental animals. This is a fair observation; in future research, it would be advisable to perform appropriate determinations in this regard.

Point 7: For all 7 figures, I recommend that the writing under each batch be more legible.

Response 7:

Point 8: English, although quite good, could be improved.

Response 8: The manuscript was subject to linguistic revision.

Point 9: On line 184 there is a parenthesis marked in yellow, immediately after reference 46.

Response 9:  Thank you very much for your attention, this mistake has now been delete.

Round 2

Reviewer 1 Report

I thank the authors for having addressed my previous comments, I'm fine with the modifications made to the manuscript. However, I suggest the following minor changes:

* Tables 1 and 2. Use the dot before decimal values instead of commas. In Table 1, the use of the terms day 1, day 7, day 14 and day 28 is recommended for the first row

* Figures. I suggest combining Figures 1-4 into a single Figure (Fig. 1) related to behavioural tests. I also suggest grouping assessments in serum and in prefrontal cortex (AChE, BChE, IL-1beta and IL-6) into two separate figures (i.e. grouping Fig. 4a, 5a, 6a and 7a into Fig. 2; and Fig. 4b, 5b, 6b, and 7b into Fig. 3). The lettering is still difficult to read in the majority of the panels. Please improve the quality of images

Author Response

Response to Reviewer 1 Comments

I thank the authors for having addressed my previous comments, I'm fine with the modifications made to the manuscript. However, I suggest the following minor changes:

Point 1: Tables 1 and 2. Use the dot before decimal values instead of commas. In Table 1, the use of the terms day 1, day 7, day 14 and day 28 is recommended for the first row.

Response 1: We would like to thank the Reviewer for suggestion, which helped us improve the quality of our manuscript. We have made changes to the tables in line with the Reviewer's suggestions.

Point 2: Figures. I suggest combining Figures 1-4 into a single Figure (Fig. 1) related to behavioural tests. I also suggest grouping assessments in serum and in prefrontal cortex (AChE, BChE, IL-1beta and IL-6) into two separate figures (i.e. grouping Fig. 4a, 5a, 6a and 7a into Fig. 2; and Fig. 4b, 5b, 6b, and 7b into Fig. 3). The lettering is still difficult to read in the majority of the panels. Please improve the quality of images.

Response 2: Due to the fact that 3 behavioral tests are presented in 4 figures (Fig. 2a and 2b - the spontaneous alternation behavior and spatial memory by Y-maze test), we decided not to make changes in the presentation of the results, because, in our opinion, the presentation of the results on one figure would be less legible. As suggested by the Reviewer, we regrouped figures 4-7.We improved the quality of the subtitles in all figures.
